# Comparing pedestrian safety between electric and internal combustion engine vehicles

Zia Wadud 

Vehicle electrification has become a major strategy for mitigating transport carbon emissions. Concerns have been raised about electric vehicle's safety impacts due to their quieter driving and heavier weight compared to conventional internal combustion engine vehicles. This research utilizes Great Britain's STATS19 road safety database to understand the pedestrian safety implications of electric vehicles. We show that the pedestrians are no more likely to collide with a fully (battery) electric vehicle compared to a conventional vehicle. In case of a collision, pedestrians are no more likely to be severely injured by an electric vehicle either. Hybrid electric vehicles have a higher pedestrian casualty rate – possibly related to their driving pattern – but the associated pedestrian injuries are less severe than those with internal combustion engine vehicles.

Globally, the transport sector is responsible for around one-fourths of anthropogenic greenhouse gas (GHG) emissions[1]. In many high-income countries, such as the UK and the US, transport is often the largest emitting end-use sector, yet it also has the slowest rate of emission reduction[2,3]. Decarbonising transport faces considerable challenges because of its perceived association with economic development, heavy dependence on fossil fuels, deeply entrenched car culture, and the large number of vehicles in operation in the world—exceeding 1.5 billion[4] —that require decarbonisation. Within this context, electrification has emerged as a central strategy for mitigating GHG emissions, particularly for light-duty passenger vehicles. Many Western countries are enacting policies to encourage electric vehicle (EV) sales and some have set ambitious targets of phasing-out only fossil-fuel-based internal combustion engine vehicles (ICEVs); e.g., the UK has banned the sale of new petrol or diesel ICE passenger vehicles from 2030[5]. Such concerted efforts have resulted in a significant increase in global EV sales in recent years.

While it is accepted that EVs can reduce carbon emissions compared to similar ICEVs – even when the electricity grid is not fully decarbonised[6] – the broader sustainability impacts of electric vehicles are still not well understood. The goals of sustainable mobility include reducing crashes, injuries and fatalities within the transport system, reducing greenhouse gases to mitigate climate change, reducing criteria air pollutants and improving public health, reducing noise pollution and improving affordability and equity of access[7]. Among these, improving road safety by reducing crashes, injuries and fatalities is a critical goal, since globally around 1.19 million people are killed and 30–50 million severely injured every year from road crashes[8]. Yet, road safety implications of EVs have received little attention from the academic community. Even those studies that exist sometimes do not agree in their findings[9,10]. Also, most of these studies focus on hybrid electric vehicles (HEVs) or HEVs and EVs together[9–11].

There are two major ways for EVs (and HEVs) to affect the safety of various types of road users compared to ICEVs. Firstly, electric vehicles could result in a larger or smaller number of collisions compared to ICEVs. Collisions could be higher due to the near-quiet driving of EVs (and HEVs) with very little noise coming from their electric motors, especially at low speeds[12]. This is especially important for vulnerable road users such as pedestrians and cyclists, who are more common in cities and are used to auditory cues from moving ICEVs. On the other hand, the EV fleet on the road is substantially younger compared to the ICEV fleet, as such a greater share of the EV fleet likely has advanced safety features like collision-avoiding systems. This could potentially reduce the number of collisions and casualties involving EVs.

Secondly, once a crash occurs, EVs – because of their substantially higher weight due to the heavy batteries – could affect the severity of the injuries of the affected persons differently compared to ICEVs. For

Institute for Transport Studies, and School of Chemical and Process Engineering, University of Leeds, Leeds, UK. ✉e-mail: Z.Wadud@leeds.ac.uk

example, an EV hitting a pedestrian would likely cause a more serious injury compared to a similar type of ICEV travelling at the same speed. Similarly, occupants of an ICEV could have more serious injuries if they are hit by a similar sized EV, which would likely be heavier. At the same time, the occupants of the EV will likely suffer a less serious injury. There is a substantial literature which shows that in a crash between two different weighted vehicles, occupants of the heavier vehicle tend to suffer relatively less serious injuries[13]. Once again, active safety features (e.g., collision avoidance), which are likely more prevalent in EVs than ICEVs could reduce the severity of injuries (e.g., automatic braking could mean that the speed is lowered by the time the impact occurs). EVs also could be different from HEVs due to their heavier weight and differences in driving patterns and environment.

This study focuses on one of the most vulnerable groups of road users – pedestrians – as 274,000 pedestrians are killed annually in road crashes around the world, which is around 23% of all road traffic-related fatalities[8]. In the UK, pedestrian fatalities and serious injuries are 25% and 22% of all road collision-related fatalities and serious injuries respectively[14,15]. Therefore, we will investigate two key research questions related to pedestrians in this study:

RQ1) whether a pedestrian's risk of collision with an EV is larger than that with an ICEV, and

RQ2) in case of a collision, whether pedestrians are likely to be more severely injured if the hitting vehicle is an EV instead of an ICEV.

We use data from Great Britain's road safety database STATS19[16,17] to answer these research questions. In order to keep the problem tractable, we narrow the focus to only those pedestrian collisions that involved cars, taxis, or private hire vehicles (PHVs). See Methods for the details of the data and analysis techniques used.

## Results

### Are EVs associated with more pedestrian casualties than ICEVs?

We use STATS19 summary data tables from 2014 to 2023 by the Department for Transport[16] to study pedestrian casualty risks. Between 2014 and 2023, there were 1,559,048 casualties of various road user types from 1,195,700 road traffic collisions in Great Britain. Of these, 210,360 were pedestrian casualties. 167,092 of these pedestrians were hit by cars (including SUVs), taxis or PHVs. Of these, HEVs were responsible for 7946 pedestrian casualties (4.76%), while electric vehicle were responsible for 1150 casualties (0.69%). These casualty statistics combine both types of injuries (slight and serious) and fatalities.

Figure 1 presents the number of pedestrian casualties arising from car, taxi and PHV collisions, the number of those vehicle propulsion types on-road and miles driven by them (along with miles walked by pedestrians), and resulting casualty rates (= casualties/miles driven) per billion miles of exposure over the last ten years. Pedestrian casualties have been decreasing overall and for ICEVs, indicating a general improvement in pedestrian safety in Great Britain (Fig. 1 panel a). However, total casualties are increasing over time for EVs and HEVs (panel a), as the EV and HEV fleet and associated traffic volume have started to grow (panels b and c) from a very small base. EV and HEV numbers and vehicle miles are still substantially small compared to the ICEV fleet. The noticeable drops in pedestrian casualties (panel a) for ICEVs and HEVs (and therefore all vehicles as ICEVs vastly outnumber EVs or HEVs) and subsequent increases are a result of the disruptions and subsequent recovery due to the COVID-19 pandemic and associated policies, which caused the traffic activities to fall substantially (panel c). Interestingly, EV traffic activities continued to grow during

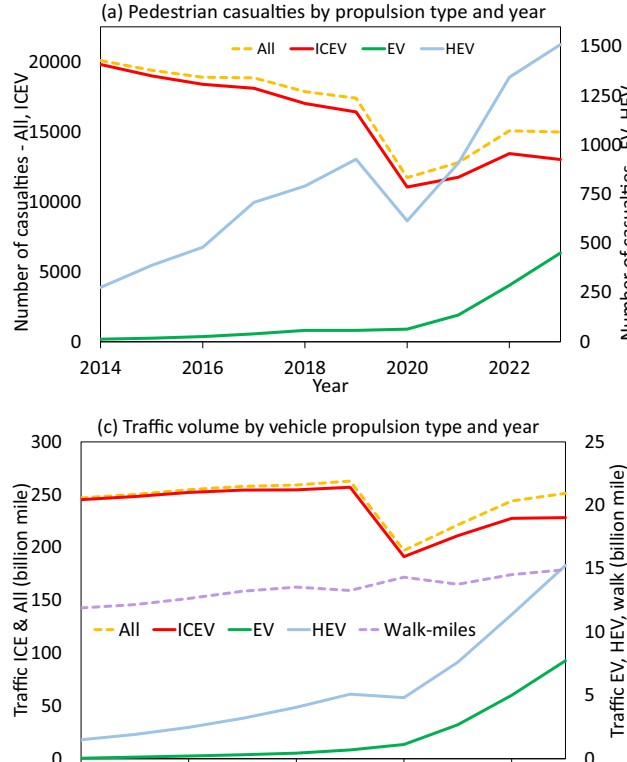
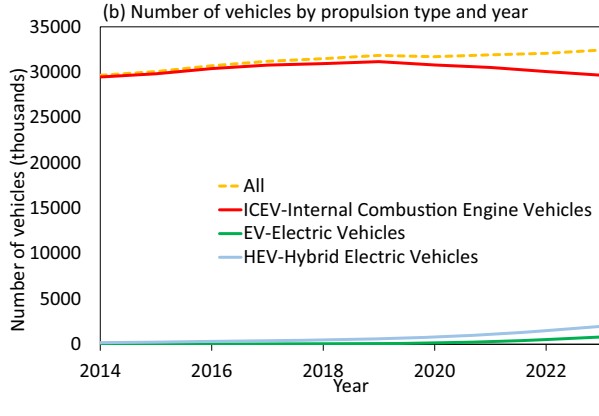
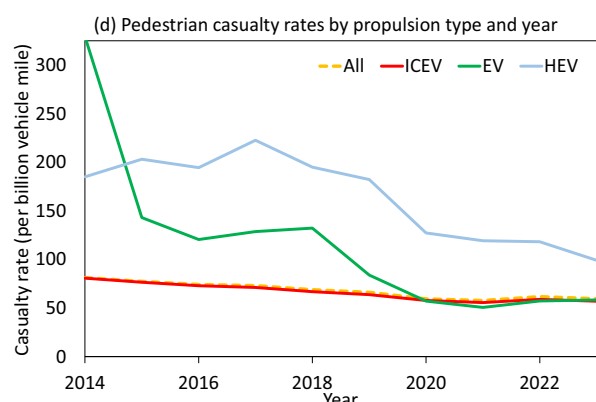

**Fig. 1 | Pedestrian casualties, car fleet composition, traffic volume and casualty rates by vehicle propulsion technology and year. a** Shows that the number of casualties is decreasing overall and for ICEVs, with some disruptions due to COVID19. However, casualties for EVs and HEVs are increasing due to their increased share in the fleet, (**b**). **c** Shows how car traffic volume, expressed in vehicle-miles, has changed in the last 10 years, with EV and HEV shares increasing. **d** Shows the trends in pedestrian casualty rates per billion vehicle-miles. EV and HEV casualties appear to have peaked, and for EVs the rate appears to have stabilized around the same rate as ICEVs. HEV casualty rates are also falling. Background data are provided as a Source Data file.

**Table 1 | Pedestrian casualty rates and casualty rate ratios for different propulsion and propulsion pair types (EV-Electric Vehicles; HEV-Hybrid Electric Vehicles; ICEV-Internal Combustion Engine Vehicles)**

| Year | Casualty rate of | | | Casualty rate ratio between | | |
|---|---|---|---|---|---|---|
| | EV (i) | HEV (ii) | ICEV (iii) | HEV-EV (ii ÷ i) | EV-ICEV (i ÷ iii) | HEV-ICEV (ii ÷ iii) |
| 2014 | 329.08 | 185.09 | 80.71 | 0.562 | 4.077*** | 2.293*** |
| 2015 | 143.16 | 203.12 | 76.50 | 1.419 | 1.871*** | 2.655*** |
| 2016 | 120.60 | 194.44 | 72.95 | 1.612** | 1.653** | 2.666*** |
| 2017 | 128.73 | 222.60 | 71.18 | 1.729*** | 1.809*** | 3.127*** |
| 2018 | 132.25 | 194.95 | 66.85 | 1.474*** | 1.978*** | 2.916*** |
| 2019 | 83.92 | 182.26 | 63.85 | 2.172*** | 1.314 | 2.855*** |
| 2020 | 57.28 | 127.32 | 57.75 | 2.223*** | 0.992 | 2.204*** |
| 2021 | 50.67 | 119.18 | 55.65 | 2.352*** | 0.910 | 2.142*** |
| 2022 | 57.37 | 118.25 | 59.05 | 2.061*** | 0.971 | 2.003*** |
| 2023 | 58.35 | 99.03 | 57.03 | 1.697*** | 1.023 | 1.736*** |
| 2019–2023 | 57.82 | 120.14 | 58.88 | P 2.078*** NB 2.195*** | P 0.982 NB 0.989 | P 2.040*** NB 2.040*** |

**, *** statistically significantly different from 1 at 95%, 99% confidence level. P – Poisson distribution; NB – Negative binomial estimation, considering overdispersion of data.

**Table 2 | Selected parameter estimates for binary logit regression for pedestrian injury severity (full model results are available in Supplementary Data 1)**

| Explanatory variables | Estimate | Explanatory variables | Estimate |
|---|---|---|---|
| EV (base=ICEV) | −0.021 | Day-time (base = weekday morn. peak) | |
| HEV (base=ICEV) | −0.261*** | Weekend super off peak | 0.760*** |
| Large SUV (base = other cars) | 0.148*** | Weekday super off-peak | 0.692*** |
| Age band of casualty (base = 0–5 years) | | Light conditions (base = daylight) | |
| 46–55 years | 0.214*** | Darkness - lights unlit | 0.433*** |
| 56–65 years | 0.492*** | Darkness - no lighting | 0.330*** |
| 66–75 years | 0.848*** | Road surface conditions (base=dry) | |
| >75 years | 1.200*** | Wet or damp | 0.160*** |
| Sex of casualty: Female (base = male) | −0.069*** | Age band of vehicle (base=0-3 years) | |
| Sex of driver: Female (base = male) | −0.101*** | 11-15 years | 0.069* |
| Age band of driver (base = 26–35 years) | | 16-25 years | 0.114** |
| 16–20 years | 0.149** | | |
| 21–25 years | 0.169*** | Model diagnostics | |
| Speed limit of the road (base = 20 mph) | | N | 53090 |
| 30 mph | 0.106*** | Log-likelihood (null) | −31585.8 |
| 40 mph | 0.570*** | Log likelihood (model) | −29280.5 |
| 50 mph | 0.932*** | Pseudo R² | 0.0730 |
| 60 mph | 0.622*** | AIC | 58757.0 |
| 70 mph | 1.427*** | BIC | 59627.2 |

***, **, * statistically significant at 99%, 95% and 90% confidence level, after Benjamini-Hochberg corrections.

this period (panel c), primarily due to the continued growth in EV numbers; as such EV-related pedestrian casualties continued to increase during the pandemic period (panel a). Pedestrian casualty rates per billion miles of car exposure for all vehicles and ICEVs have been slowly and steadily decreasing over these 10 years (panel d). The casualty rates are less stable during the earlier years for EVs and HEVs, which is not unusual during the introduction of a new technology. Qualitatively, the casualty rate for EVs appears to be decreasing and seems to have stabilised recently, at around the same rate as ICEVs (panel d). A similar downward trend can be observed for HEVs too (panel d).

In order to answer RQ1, we use the summary casualty rates statistics from Fig. 1 and Table 1. We compare the pairwise casualty rate ratios for EVs, HEVs and ICEVs during 2019-2023, in order to avoid the earlier volatile years; 2019 is also when EV sales started to grow rapidly after rather slow growth during the earlier years (see Methods). Around 996 (1.38%) and 5303 (7.36%) pedestrian casualties (out of 71,979) could be attributed to EVs and HEVs respectively during 2019–2023. The average pedestrian casualty rates per billion miles of driving during this period were 57.82 for EVs, 120.14 for HEVs and 58.88 for ICEVs. Casualty rate ratios during those years for EV-ICEV, HEV-ICEV and HEV-EV were 0.99 (95% CI 0.91–1.08), 2.04 (1.73–2.40) and 2.15 (1.71–2.71), respectively using a Negative Binomial model (see Methods). Therefore, the casualty rates for EVs and ICEVs are *statistically not different* from each other (casualty rate ratios are not different from 1) for these five years. Using the more conservative Poisson model also, casualty rate ratios for EVs and ICEVs are not statistically different from unity, both collectively and for each individual year. However, throughout all those years, HEVs show a higher casualty rate, with casualty rate ratios larger than 1 with respect to both EVs and ICEVs, for nearly all individual years and the recent five years collectively.

**Do pedestrians suffer more severe injuries in collisions with EVs?**
RQ2 is answered using the 'casualty' micro-dataset for 2019–2022 (see Methods). We have created two categories of injury severity: slight, and severe (which includes serious injuries and fatalities, see Methods). Table 2 presents selected parameter estimates from the preferred binary logistic regression model for casualty severity, while full estimation results are available in Supplementary Data 1. A statistically significant positive parameter estimate in Table 2 would suggest that the corresponding variable increases the likelihood that a pedestrian

casualty will fall in the severe (serious or fatal) category as opposed to the slight category.

The key parameters of interest are for the indicator variables for EVs and HEVs. Parameter estimate for EVs is statistically insignificant, even at a *conservative* 90% confidence level, indicating pedestrians hit by EVs are *no more likely* to suffer a severe (serious or fatal) injury compared to ICEVs, our base vehicle type. It suggests that the potentially increased danger due to the EV's heavier weight likely gets countered by the presence of active safety measures generally available in the EV fleet, as suggested earlier. While this cannot be proven conclusively using the current data, HEV's parameter estimate sheds some insight. HEVs show a negative and statistically significant association with the severity of injuries. Given HEVs are also a newer fleet compared to the ICEV fleet on the road, a larger share of HEVs likely have better active safety measures, making them safer for pedestrians in case of a collision. This benefit is possibly even larger for EVs (which are more recent than HEVs and likely more generously equipped), but their substantially increased weight possibly acts against it, yielding the statistical insignificance of the corresponding EV parameter estimate.

Other parameter estimates show expected associations. Clearly, collisions with SUVs increase the likelihood of a serious or fatal pedestrian injury. There is an abundance of evidence on SUV's and large vehicle's adverse impact on road safety because of their heavier weight and (for SUVs) their body shape[18–20]. Collisions with older vehicles are more likely to be more harmful, potentially due to their lower safety standards. Similar findings were reported in Norway, too[21].

Female drivers are less likely to cause severe pedestrian injuries, but younger drivers tend to have the opposite effect, both of these findings are supported in literature[22,23]. Older pedestrians are more

likely to suffer severe injuries – as found before[24]. Male pedestrians also have a higher likelihood of experiencing severe injuries, another finding supported by literature[25].

While the dataset does not contain information on the speed at which the vehicle was driving, the effect of speed limits on the roads where the collisions took place can shed some light. As speed limit on the roads increase, so does the likelihood of a serious or fatal injury. This is again a well-established finding[26–28]. We could not include urban and rural areas separately in the model, as speed limits are highly correlated with urban/rural indicator (see Methods). However, the results imply that the risks of more severe collisions in cities would be less compared to those in rural areas, given that urban roads will more often than not have lower speed limits. Various temporal, road surface and light conditions, vehicle manoeuvres and junction types affect the likelihood of severity of pedestrian injuries, and follow expected patterns (e.g., crashes during darkness, super off-peak periods and on wet road surfaces leading to a higher likelihood of severe injuries).

## Discussion

Our results on car or taxi-related pedestrian casualty rates and pedestrian casualty severities show that: 1) casualty rates for EVs are no worse than those for ICEVs, and 2) once a collision occurs, the severity of injuries to pedestrians due to EVs is not different from that for ICEVs. Together, they show that EVs – specifically post-2019 – are no more dangerous than ICEVs for pedestrian safety. As such, the current pattern of adoption and use of EVs do not adversely affect the achievement of the important transport sustainability goal of reduced road crashes, injuries, and fatalities.

Road user safety has always been an active research field in transport studies, although research on on-road safety implications of electric and hybrid electric vehicles is rather limited. The earliest among these found that HEVs in the USA were twice as likely to be involved in a collision involving pedestrians during 2000–2007[11]. A later update reinforced the finding for HEVs, but the odds ratio narrowed to 1.35[29]. In the UK, an early study[10] showed that EV/HEVs were no more likely to be involved in a collision with pedestrians compared to ICEVs during 2005–2008, a finding challenged recently using crash data from 2013–2017[9]. Although both studies include EVs in their calculations, there were very few EVs in the UK during the study period (e.g., even in 2017, only 0.12% of the car vehicle fleet consisted of EVs, whereas the HEV share was ten times higher at 1.23%); so in essence, the results were for HEVs. As such, pedestrian collision rates involving fully battery electric vehicles (EVs) are not available in the literature to directly compare our results against.

Other studies found that the proportion of EV (or HEV) crashes involving pedestrians among all EV (or HEV) crashes was larger compared to such proportion for ICEVs in Norway[30] during 2011–2018 and the UK[10] during 2005–2008, potentially because EVs (or HEVs) were driven more in urban areas. Yet, in Virginia, USA, there was no such difference between EVs and ICEVs between 2017–2023[31]. Also, the share of serious or fatal crashes for EVs in Spain was no different than that share for ICEVs[32]. None of these studies, however, directly provides causality rates (casualties per billion vehicle miles) as in the first objective of this study.

Because of concerns about the quiet nature of electric driving, all new models of EVs and HEVs in the UK were regulated to add an acoustic vehicle alert system (AVAS) to caution nearby road users during low-speed (< 12 mph) driving from July 2019. This appears to coincide with a fall in casualty rates of EVs (Fig. 1 panel d). Average casualty rates for HEVs during 2014-2018 and 2019-2023 were 201.6 and 120.1, respectively (Table 3), which are statistically different at a 99% confidence level. A similar statistically significant drop (137.2 to 57.8) in casualty rates is observed for EVs, too (but this needs to be interpreted *cautiously*, given the early year estimates were quite

**Table 3 | Comparison of pre-and post-AVAS pedestrian casualty rates and rate ratios (EV-Electric Vehicles; HEV-Hybrid Electric Vehicles; ICEV-Internal Combustion Engine Vehicles)**

|  | EV | HEV | ICEV |
|---|---|---|---|
| Casualty rate - pre AVAS (2014-2018) | 137.19 | 201.63 | 73.57 |
| Casualty rate - post AVAS (2019-2023) | 57.82 | 120.14 | 58.88 |
| Casualty rate ratio – post/pre AVAS periods | P 0.421*** | P 0.596*** | P 0.800*** |
|  | NB 0.421*** | NB 0.643*** | NB 0.797*** |

*** statistically significantly different from 1 at 99% confidence level. *P* Poisson, *NB* Negative Binomial.

volatile). It is important to note that the overall pedestrian safety has been improving over the decade and as such, casualty rates between the pre- and post-AVAS five-year periods have fallen for our counterfactual ICEVs, too. However, the rate drop is substantially smaller for ICEVs (73.6 to 58.9) compared to HEVs or EVs. This indicates that the drops for EVs and HEVs were larger than the level expected from the gradual improvements in pedestrian safety. EV numbers jumped from 292,000 at the end of 2021 to 806,000 at the end of 2023, accompanied by a rapidly increasing number of new EV models since 2019 (all new models required AVAS), which suggests that a large share of EVs had AVAS post 2019. These together indicate that the AVAS regulation likely contributed to reducing pedestrian casualty rates of EVs and HEVs. While robust conclusions cannot be drawn here, future studies should investigate changes in casualty rates by speed categories when relevant data become available: a larger fall in low-speed casualties for EVs compared to ICEVs will lend further support to a causal association.

HEVs clearly have a larger casualty rate compared to both ICEVs and EVs, as found here and in previous studies[9]. This higher pedestrian casualty rate is intriguing. In addition to the quietness of the HEVs (a substantial share of HEVs on-road are still pre-AVAS vehicles), two additional factors potentially contribute to HEV's poor performance in the UK. Firstly, HEVs are very popular in the taxi and PHV fleet in the UK due to their low running costs compared to petrol vehicles and better emission performances compared to previously popular diesel vehicles, which would have been incompatible with clean air zone regulations in many large cities. These HEVs are driven on an average of three to four times as much as a typical private car, meaning our exposure vehicle-miles for HEVs may have been underestimated, so casualty rates were likely overestimated. Secondly, most of these taxi and PHV miles are also driven in urban areas, where they are exposed to pedestrians more. A recent study[9] indeed reported that the EV-HEV fleet (overwhelmingly dominated by HEVs) had a higher casualty rate in urban areas compared to ICEVs during 2013-2017. As such, the issue may be less a result of whether the vehicles are HEV or ICEV, rather than where and how the vehicles are driven. Investigating such potential confounding factors (e.g., driver age or pedestrian density of areas where the vehicles are driven) is an important avenue for future research – both for HEVs and EVs. Obtaining exposure data for such detailed analysis is a potential challenge.

The role of active safety technologies on pedestrian and overall safety performance is another area that requires immediate attention. This follows our hypothesis that the presence of active safety technology in EVs may have made these vehicles as safe as the existing ICEV vehicle fleet. Quantifying this impact of technology for all road users and vehicle types and subsequent policy or regulatory measures could help improve the safety and sustainability of all passenger cars – EVs or not – in future. Especially, it is still not implausible that, like for like – pedestrian injuries sustained in ICEV collisions could be less severe

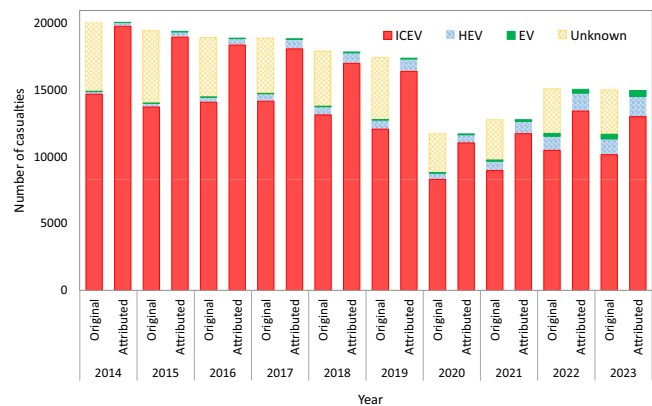

**Fig. 2 | Original estimates for casualties by propulsion types and estimates after attributing the unknown casualties.** Background data are provided as a Source Data file.

than those in EV collisions if both vehicles had the same safety features. Future studies should also investigate the effects of EVs in vehicle-to-vehicle collisions to understand the potentially beneficial effects on EV occupants in order to ascertain the overall impacts of these vehicles on different types of users.

While our focus has been on the safety implications of conventional four-wheeled passenger vehicles on pedestrians, electrification is happening in two-wheeler and three-wheeler vehicle segments, too. Especially, indigenously electrified cycle-rickshaws, or locally assembled and unregulated electric autorickshaws have become a popular form of paratransit in the cities of some low-income countries[7]. Anecdotal evidence suggests that these novelty EVs can be a safety risk for both pedestrians and users and require further attention from researchers and policymakers.

## Methods

In order to investigate the two RQs, we utilize Great Britain's STATS19 road safety database, which contains every officially reported collision in Great Britain. It consists of three separate micro datasets for every collision: collision characteristics (e.g., road conditions, weather conditions, location, etc.), colliding vehicle characteristics (vehicle make, model, fuel type, driver age, gender, etc.), and casualty characteristics (age, gender, etc.). We use three propulsion types for both RQs in this study: EVs, where we group battery electric vehicles and fuel cell vehicles (very few) together, hybrid electric vehicles (HEVs) include petrol and diesel hybrid electric vehicles, diesel electric vehicles and plug-in hybrid electric vehicles; ICEVs consists of the rest – which are mostly diesel and petrol vehicles, but can include a few natural or petroleum gas vehicles, too.

In order to answer RQ1, we use summary tables (known as incidence tables in Epidemiology) of the total number of pedestrian casualties due to collisions with cars or taxis, by the propulsion type of the vehicles. These summary casualty statistics have already been derived by the Department for Transport[16] from the STATS19 microdata (Table RAS0507). Unfortunately, a non-trivial number of vehicles do not have any information on the type of propulsion. We assume that these collisions are also distributed similarly to the known propulsion types, and allocate those to the three propulsion types on a proportional basis. Figure 2 shows the original and redistributed casualty numbers.

Despite the recent increases in EV sales share, the UK on-road car fleet is still dominated by ICEVs, as shown in Fig. 1(b), with data from the Department for Transport[33]. In order to compare the risks associated with different propulsion types, it is important to control the number of pedestrian casualties with respect to pedestrians' exposure to these different types of cars. We do so by using the number of miles travelled by different car propulsion types in the collision year (Fig. 1(c)). The Department for Transport[34] estimates total car traffic activities for each

year; these are distributed to fuel types by using the corresponding share of cars in car parc. This has an inherent assumption that cars with different propulsion types (in this context, EVs, HEVs, and ICEVs) drive the same distances on average. Given that the early EVs were substantially constrained by range, the mileage per EV has been increasing over time due to improvements in battery technology, and the number of EV casualties is small in the overall casualty data, it is especially important to account for the evolving EV miles over time. We utilize information from the RAC Foundation's Green Fleet Index[35], which utilizes the Ministry of Transport (MoT) annual fitness tests, which record odometer data for every UK vehicle older than 3 years, to provide average EV miles for each year. Using miles (instead of vehicle numbers) to estimate exposure allows us to control for the rather variable amounts of miles driven in different years, especially due to the disruptions during the COVID-19 pandemic and subsequent recovery, when mileage was affected but not vehicle numbers (Fig. 1. panels b and c). Our casualty rate is thus calculated as the number of pedestrian casualties per billion miles of travel for the three vehicle propulsion types.

We use pairwise casualty rate ratios (also known as incidence rate ratios in Epidemiology) for EV-ICEV, HEV-ICEV, and EV-HEV to compare the risks associated with each vehicle type. An incidence rate ratio of 1 would indicate that the casualty rates for the two vehicle types compared are similar. Since the number of casualties is a count variable, Poisson and Negative binomial distributions are commonly used in estimating the rate ratios and corresponding confidence intervals, while the exposure variables are the corresponding vehicle-miles for the respective propulsion types. Poisson distribution assumes that the variance is equal to the mean, and the confidence interval of the estimated rate ratios will be narrower than that for the Negative Binomial distribution, which can accommodate overdispersion and is not uncommon for road traffic collisions. However, the Negative Binomial distribution requires estimation of an additional overdispersion parameter, which is not possible for the pairwise summary data for individual years in Table 1. As such for yearly rate ratios, we are constrained to using the Poisson distribution. For the combined statistics for multiple years – our primary interest in the 5 years post 2019 (bottom row of Table 1) – we could estimate the confidence intervals using the Negative Binomial distribution. For pre-post comparison in Table 3 we also provide estimates using both Poisson and Negative Binomial distributions. Both models point to the same conclusions for these multi-year comparison cases; however, the overdispersion parameters were statistically significant, indicating that the Negative Binomial results are more appropriate. As such our main conclusions are based on Negative Binomial models.

While we present the data for the last 10 years – 2014 to 2023 in Table 1, the number of pedestrians hit by EVs was quite small pre-2019, as was the number of EVs and the amount of driving done by those early EVs, which were often constrained by range in a single charge. Figure 1 clearly shows the rather fluctuating collision rates in earlier years, which is an indication that the system probably had not reached a stable equilibrium. EVs reached parity with ICEVs in the UK in terms of annual driving per car in 2019. Also, EVs and HEVs were asked to introduce AVAS in 2019, imparting some important discontinuity in the data. As such, statistical inferences on casualty rate ratios between different propulsion groups are made on the basis of the most recent 5 years of data (2019-2023) to answer RQ1.

RQ2 utilizes individual-level casualty data for 2019–2022 (Department for Transport[17]) as 2023 microdata were unavailable when the work started. We focus again on collisions of pedestrians with cars and taxis, and restrict the sample to one vehicle collisions only, since it is difficult to ascertain which vehicle may have hit the pedestrian for multi-vehicle collisions. Our key variable of interest for RQ2 is injury 'severity'. Injury severity has three categories in the STATS19 'casualty' dataset: slight, serious and fatal. Given the very few fatalities involving EVs, we group serious and fatal casualties together to create a 'severe' injury group, which provides us with a binary dependent variable.

The severity of an injury to pedestrians in a collision depends on a number of factors[36]: a) pedestrian characteristics (e.g. age, gender), b) driver characteristics (age, gender), c) vehicle characteristics (e.g., body type, weight, speed), d) road characteristics (e.g. speed limit, road quality), e) environmental factors (e.g., weather, time, visibility) and f) collision characteristics (e.g. vehicle movement). We combine the 'casualty' microdataset with 'vehicle' and 'collision' dataset of STATS19 via the unique identifiers in order to collect and connect as many of the relevant explanatory factors as available for our model.

Car body-type has a significant role in the severity of injuries, especially, SUVs are known to be more dangerous to pedestrians because of their higher bonnet height compared to cars[37,38]. Although STATS19's 'vehicle' dataset includes vehicle make and model, there is no official definition of SUVs, and as such, no official dataset exists with this classification. The situation is further muddled for our context, given the consumer preference for SUV-type styling of small and medium cars with rather low-powered engines in the UK. We utilised an online commercial vehicle trading website's listing of vehicles to identify SUVs and supplemented that manually by cross-checking model and make on manufacturers websites for some makes and models. Then we created a dummy variable to identify if the hitting vehicle is a large SUV (SUV styling and engine>2000 cc) or not.

Both statistical regression and data-mining or machine learning techniques have been used to understand the injury severity of collisions. Statistical methods include, among others, binary logistic or logit[39,40], binary probit[41], multinomial logit[42], ordered logit[43], ordered probit[44], generalized ordered logit[45] and generalized ordered probit models[46]. Popular data mining methods applied include random forest, support vector machine, decision tree, K-nearest neighbour cluster and neural networks[47]. Given our dependent variable, injury severity is categorical with two levels (slight and severe), we run a binary logistic regression model to understand how the predictor variables can explain the probability of an injury being severe (as opposed to slight). An introduction to the binary logistic model is available in Gujarati et al.[48], while a more detailed treatment is available in Hosmer et al.[49]. Given that some collisions can have more than one casualty, the observations are clustered at the collision level. Table 2 above presents selected parameter estimates, while Supplementary Data 1 presents the full model results.

Speed and urban/rural indicator could both be important predictors of injury severity. However, these two variables are highly correlated as rural areas generally have a larger speed limit. This multicollinearity meant we had to opt for only one of these variables. We tested three separate models with speed only, urban/rural indicator only and both as predictors. The model with speed only performed the best in terms of goodness of it. Supplementary Data 2 presents the summary statistics for the explanatory variables that entered the regression model.

Although we are primarily interested in the role of EVs and HEVs on pedestrian injury severity, we also commented on the role of other contributory factors. As such, Table 2 presents results for corrected *p*-values using Benjamini-Hocheberg's[50] false discovery rate correction to avoid accidental identification of a significant relationship in the presence of multiple hypothesis tests. Results for an alternate binary probit model are also presented in Supplementary Data 1. Results of both models agree and the binary logistic model exhibits slightly better goodness-of-fit.

## Reporting summary
Further information on research design is available in the Nature Portfolio Reporting Summary linked to this article.

## Data availability
All STATS19 micro and summary data, and transport indicators such as vehicle miles and vehicle stock are publicly available from the Department for Transport, as described in the Methods section. SUV indicator data is confidential. Source data for Fig. 1 and Fig. 2 are provided with this paper.

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

## Acknowledgements
The author thanks Tim Chatterton and Malcolm Morgan for sharing some of the data, Aashaz Zia for his support in data processing, and Samantha Jamson for her useful comments on the first draft.

## Competing interests
ZW declares no competing interests.
