## [Transparent Peer Review file · Nature Communications]

Comparing pedestrian safety between electric and internal combustion engine vehicles

Corresponding Author: Professor Zia Wadud

Version 0:

Reviewer comments:

Reviewer #1

(Remarks to the Author)

This study examines the safety of Electric Vehicles (EVs) and Hybrid Vehicles (HEVs) for pedestrians, compared to vehicles with only an internal combustion engine (ICEVs), in the UK. It found that in recent years, EVs are at least as safe as ICEVs. HEVs had pedestrians higher casualty rates, but caused less severe injuries than ICEVs. Another key finding was that an auditory vehicle alert system in EVs and HEVs, mandated on cars from 2019 has reduced pedestrian casualty rates.

I believe this to be an important topic and, as the author states, there is a need for more research in this area. As a results I think this research could be quite significant.

However, I think there are some flaws that need to be addressed or acknowledged prior to publication.

One of these relates to the inclusion of Taxis and PHVs in the analysis relating to RQ1, and their potential unequal influence on the denominator (vehicles miles) for the different types of vehicles. While this is acknowledged in the discussion as a potential explanatory factor for the high rate of pedestrian crashes for HEVs, I think the study would have been improved by excluding Taxis and PHVs from the analysis for RQ1.

I believe another flaw is the manner in which vehicles of unknown propulsion type were all assigned to ICEVs. That most vehicles with known propulsion are ICEVs does not justify this decision. While it may produce a casualty rate that is closer to the true rate for ICEVs, it then underestimates the rate for EVs and HEVs. Either the unknowns need to be excluded, or they need to be assigned to all vehicle types considered based the proportions with known propulsion type. Judging by the results provided when the unknowns were excluded, this changes the conclusions, as EVs would have a significantly higher casualty rate than ICEVs.

I think the conclusion that EVs are at least as safe as ICEVs should really be revised to being that EVs with AVAS are at least as safe to more accurately reflect the results, and ensure the result is not misapplied to countries where AVAS is not required.

The crash data source is the STATS19 database of police reported crashes in the UK. I am unfamiliar with any particular limitations of this data, though I am aware that police reported crash data may have some coding errors, missing data, and for some biases, especially for variables that are self-reported by the driver. However, it is appropriate for use in this study, provided the problem posed by the missing data on vehicle propulsion is handled well.

The paper refers to the data used in answer RQ2 as the "micro-dataset". It is unclear what is source is, though I assume it is a subset of the STATS19 data. It is also unclear why the data was restricted to 2019-2022.

The other main data used is vehicle miles travelled reported by the Department for Transport. I note that this data source could use an exact reference (e.g. was it from a report or series of reports?). The analysis assumes that equal vehicle miles are travelled by all vehicle types (EV, HEV, ICEV). This may or may not be true. I would like to see this study include some more supporting evidence (e.g. references) that support this assumption. There is a line in the Methods section, paragraph 4, that states "EVs reached parity with ICEVs in the UK in terms of annual driving per car in 2019", but there is no reference. I note that different data on EV mileage is presented in the supporting information. Again, a clear reference and explanation of

how that vehicle mileage was estimated would be useful. I think it is also a bit odd to have this as supporting information. If the author thinks the mileage data is superior, they should use it in the study.

I find the method used to answer RQ3 unusual. When considering a vehicle safety technology that was mandated from a certain point in time, the analysis would usually look at the year of manufacture of the vehicles, and look at those before and after. To use calendar year of crash is unusual. This would mean that there would be many EVs and HEVs involved in pedestrian crashes in the Post-AVAS category that do not have AVAS. This problem is mitigated somewhat by the low number of EVs, and their rapid increase in sales. However, I don't see why the analysis used calendar year of crash rather than year of manufacture. Vehicle mileage could still be taken into account.

I note that exact n and p values were not reported for all results, as is requested in the author guide.

With regard to references, I found that there were many statements made or statistics quoted that lacked an appropriate reference to back them up. This includes.

Main-Paragraph 1, the first two sentences.

Main-Paragraph 1, second to last sentence

Main – Paragraph 3, second to last sentence

Main, do pedestrians suffer more..., Paragraph 2, last sentence

If references to back up these statements cannot be provide, they should be deleted.

Reviewer #2

(Remarks to the Author)

NATCOM-Are electric vehicles more dangerous for pedestrians?

The study uses STATS19 to analyze pedestrian casualties related EVs, HEVs, and ICEVs. The findings show no significant difference in collision rates or injury severity between EVs and ICEVs. However, HEVs have a higher pedestrian casualty rate than ICEVs. The paper claims the introduction of AVAS in 2019 correlates with reduced casualty rates for both EVs and HEVs, suggesting policy effectiveness. The study concludes that EVs are at least as safe as ICEVs for pedestrians.

My key concerns include uncontrolled variables and insufficient separation of low-speed collisions to isolate AVAS impacts. In addition, AVAS policy analysis is questioned due to phased implementation (2019–2021) and the concurrent rollout of Autonomous Emergency Braking systems, potentially conflating causality. In summary, I believe comments 7-9 must be addressed for publication.

Major comments:

1. A few uncontrolled variables limit the extent of conclusions made from the available data. For example, it is implicitly assumed that the EV and ICEV drivers have the same driving skills. If the assumption is not true, it affects the conclusions. Considering the likely scenario of EVs being more attractive to younger drivers, the possible casualty rate difference may be due to this uncontrolled variable rather than the vehicle type. Pedestrian attitude toward vehicle type, pedestrian density, vehicle's physical condition, age, and mileage are other uncontrolled factors.
2. Another uncontrolled variable that affects the reliability of the conclusions is the accident location, urban or rural areas. The prevalence of EVs and ICEVs might differ in those areas, as well as the casualty rates and severity. For example, assume the casualty severities are lower in urban areas where EVs' presence is expected to be higher. Then, the tests will show that EVs have less severe accidents, when in fact, this is a result of the location rather than the vehicle type. Without separating the dataset, the results cannot be reliably attributed to vehicle type.
3. "We could not model urban and rural areas separately, given speed limits are highly correlated with urban/rural indicator." Since the database contains the location information of the accidents, I wonder why it was impossible to model urban and rural areas separately. Without this separation, it is difficult to attribute results reliably to vehicle type.
4. In Table 1, all the rows are compared to row 1, 2014. As mentioned in the text, due to the fluctuations and deficiencies in EVs' early years, this is not a proper base. I suggest using the year 2018 or 2019. The same applies to Table 3.
5. In Table 1, most rows are different from the base, but we don't know if they are different from each other. In addition, I suggest using notched boxplots for visual statistical significance tests between the cases that are discussed or built upon in the manuscript.
6. Like comment 4, in Table 2, the statistical significance test bases are apparently not properly chosen, for example, day-time combination or speed limit. Moreover, sample sizes are not mentioned. I believe notched box plots with sample sizes would resolve the issue. It will be much easier to read and interpret the significance between every pair. I recommend adding it to the supplementary material.
7. Regarding the mandated Acoustic Vehicle Alerting System, in July 2019 it was implemented on only "new models". From July 2021, all "newly registered" models must have the alarm. Therefore, a large portion of EVs manufactured before 2019, and old models produced between 2019 and 2021 are not required to have the AVAS. As expected, the policy would take a couple of years to be fully implemented. I believe the casualty rate drop is unlikely to be related to the new policy, as the majority were not equipped with the system in the years studied in the paper, e.g., 2019-2023.
8. EVs are required to emit sound at speeds lower than 20 km/h. At higher speeds, the policy is not mandated. Therefore, to study the effects of the policy, only casualties at low speed should be compared. However, the manuscript compares all casualties before and after the implementation at all speed ranges; see Table 3. If the effect is significant only at low speeds, it is evidence of the new regulation's effectiveness. But if the effect is significant at all speed ranges, then, it is evidence

against it being related to the policy. The paper doesn't separate the low-speed cases and therefore the statistical significance test doesn't support the policy's effectiveness.

9. Autonomous Emergency Braking (AEB) with pedestrian detection was introduced two or three years before AVAS. The technology had a couple of years to show its effectiveness during the period studied in the paper. It is not speed-related and is compatible with the results. The casualty drops may be related to the technology upgrade.

10. The study has a multiple comparisons problem. When performing many statistical significance pair-wise tests, for example in Table 2, the chance of getting accidental statistical significance increases. I suggest performing a false discovery rate analysis like Benjamini-Hochberg.

Minor comments:

11. Since the novelty of the study is the investigation of EVs, which were not present in the literature before, I suggest adding a few lines explaining the motivation. I am looking for reasons why the authors thought EVs may be different from HEVs.

12. "Even those studies that exist sometimes report contradictory findings (Edward et al. 2024, Morgan et al. 2011)."

I don't see any contradiction between the two studies. Edward et al. discuss that the number of accidents per travel length is higher for EVs, while Morgan et al. focus on visually impaired pedestrians and the increased collision risk with quiet vehicles.

13. In Fig. 1, there are a few issues; it is unclear what walk-miles and ICEV-Low are (ICEV-Low is defined in the methods section. Still, I suggest defining them in captions); all the y-axes' titles are vague or partially described; the x-axis in (a) is truncated.

14. When using casualty rate and aggregated casualty rate, please explicitly define what you mean.

15. Please describe what each data point is (I guess a recorded accident?), and mention the size of the groups being statistically compared or how many samples each group has.

16. In the text, HEVs are more dangerous than EVs and ICEVs. Since this is unintuitive, it would be helpful if the authors explained the reason.

17. In Table 1, please define the abbreviations in the caption for readability. Moreover, please mention the number of samples in each group.

18. "RQ2 is answered using the 'casualty' micro-dataset for 2019-2022."

Why did you exclude 2023 for RQ2, whereas it was used for RQ1?

19. After Table 1, it would be helpful to explain how the binary logit regression model is applied.

20. I think the speed limit for AVAS is 20 km/h and not 20 mph.

21. The authors mention

"As a sensitivity analysis, we removed the missing-propulsion type casualties from ICEVs and re-estimated the ICEV casualty rates (ICEV-Low in Fig. 1 and Table 1). Even in this extreme and highly unlikely scenario, EV casualty rates are statistically not different from ICEV rates for five of the ten years (Table 1)."

And

"We also run a lower estimate for ICEV casualty rate (ICEV-low), where we do not include the unknown collisions to ICEVs, but believe it grossly underestimates ICEV casualty rates."

The exclusion makes perfect sense. Why do you think it is highly unlikely, and it grossly underestimates casualty rates?

Aren't the excluded data supposed to have the same distribution characteristics as the whole set?

22. In the car body type study, please explain the dummy variable and how you use it.

23. The abstract mentions quieter driving and heavier weight as the reasons for EV safety concerns. The AVAS study justifies the former but I don't see a reason to mention the latter in the abstract. The introduction may discuss heavier weight and its implications with proper citations.

24. Why are the authors talking about car body type and SUVs? I don't see how they relate to the paper's research questions. Can you explain why you are interested in SUVs or separating them from the rest?

Reviewer #3

(Remarks to the Author)

The manuscript addresses an important research question regarding the pedestrian safety implications of electric vehicles. While the topic is timely and the use of collision data from STATS19 is appropriate, several methodological concerns limit the reliability of the findings. The following issues should be addressed in a future iteration of the paper.

1) It is not clear how the exposure data were obtained and processed. The STATS19 database includes only collision records and provides very limited traffic volume or exposure information. The paper should clearly explain the source of traffic data and the method used to allocate exposure across vehicle propulsion types.

2) Similarly, the methodology for estimating annual miles driven by different vehicle types is inadequately described. The paper offers little information on how these figures were derived or adjusted for differences in usage patterns (e.g., taxis vs. private vehicles), which is essential for accurate rate calculations.

3) The binary logit regression model used for injury severity includes a large number of statistically insignificant variables. This overfitting may introduce bias and reduce the reliability of the estimates. A more parsimonious approach to model specification with only statistically significant is recommended.

4) The use of the Poisson distribution for computing confidence intervals may be problematic, as collision data are often over-dispersed. The Poisson model assumes the mean equals the variance, which is rarely the case in traffic safety data. Alternative methods should be considered.

5) The use of a 10-year dataset without accounting for potential structural changes over time raises concerns. Multi-year data can exhibit temporal instability, where parameter relationships shift due to evolving technologies, policies, or external factors (e.g., COVID-19). This issue is well-documented in the literature—see, for example, Mannering (2018), “Temporal instability and the analysis of highway accident data.” Addressing this concern may require segmented analysis or the inclusion of interaction terms to capture time-based variation.

6) The literature review omits key studies focusing on safety issues of similar types of vehicles.

Version 1:

Reviewer comments:

Reviewer #1

(Remarks to the Author)

I believe all my comments have been adequately addressed. Thank you.

Reviewer #2

(Remarks to the Author)

1. I see. Then, I recommend adding a statement to the limitation section mentioning that the insufficient partitioning in the source data prevented the authors from isolating such effects. I like the reader to know there are other untested hypotheses that may change the conclusions if better portioned data becomes available.

2. Thank you.

3. Thank you.

4. I see. Please add a note, maybe in the table caption to avoid confusion.

5. Same as 4.

6. I still think a different base selection better represents the data. However, as the authors mention, it doesn't change the results. I leave it to the authors' judgment.

7. The statement requires supporting data. Any further build-up on the assumption of the majority of EVs having AVAS from 2019 to 2023 must be carefully examined (and what majority, 60%? 70%? 95%?). I agree that as time passes, the percentage increases, but is it fast enough for the effect to be visible in casualty rates? I still believe that this is not the case.

Overall, I recommend a year-wise percentage study for EVs with and without AVAS.

If impossible, please add a note regarding the rollout speed limitation, as I explained in the original comment where the authors are discussing AVAS effectiveness. Please clearly mention both views and the reason why you believe it is not the case.

(Currently, the authors ignore the point and just mention that the result is not robust due to other reasons.)

8. I understand that the data is limited. But I believe that isolating low speeds is necessary for the conclusions made in the original submission. In the current version, a more direct mention of speed study is helpful (the current form is too vague).

9. Thank you.

10. Thank you.

11. Yes. But how are those important in the analysis?

12. Thank you.

13. Please make sure that the terms are properly introduced. Thank you.

14. Thank you.

15. Thank you.

16. I assume it is already explained in the text.

17.I know. I meant for readers who look at the figures and tables first. Please explain that the sample size is not available and why, where it fits in the manuscript, if not already explained.

18.I see. Please explain it in the text for the curious reader.

19.Thank you.

20.Thank you.

21.Thank you.

22.I see.

23.Ok.

24.I see.

Version 2:

Reviewer comments:

Reviewer #2

(Remarks to the Author)

I believe the manuscript meets the standards for publication.

RESPONSES TO THE REVIEWER COMMENTS

Reviewer #1 (Remarks to the Author):

This study examines the safety of Electric Vehicles (EVs) and Hybrid Vehicles (HEVs) for pedestrians, compared to vehicles with only an internal combustion engine (ICEVs), in the UK. It found that in recent years, EVs are at least as safe as ICEVs. HEVs had pedestrians higher casualty rates, but caused less severe injuries than ICEVs. Another key finding was that an auditory vehicle alert system in EVs and HEVs, mandated on cars from 2019 has reduced pedestrian casualty rates.

I believe this to be an important topic and, as the author states, there is a need for more research in this area. As a results I think this research could be quite significant.

We thank the reviewer for the positive view of the paper. Our responses to the reviewer's comments are given below in italics.

However, I think there are some flaws that need to be addressed or acknowledged prior to publication.

One of these relates to the inclusion of Taxis and PHVs in the analysis relating to RQ1, and their potential unequal influence on the denominator (vehicles miles) for the different types of vehicles. While this is acknowledged in the discussion as a potential explanatory factor for the high rate of pedestrian crashes for HEVs, I think the study would have been improved by excluding Taxis and PHVs from the analysis for RQ1.

We have used summary data (table RAS0507) from DfT. Unfortunately, taxis and PHVs are not reported separately for this to be carried out for the 10 years. A finer resolution analysis also requires a reliable estimation of vehicle-miles for taxis/PHVs for each propulsion type which are also not available.

I believe another flaw is the manner in which vehicles of unknown propulsion type were all assigned to ICEVs. That most vehicles with known propulsion are ICEVs does not justify this decision. While it may produce a casualty rate that is closer to the true rate for ICEVs, it then underestimates the rate for EVs and HEVs. Either the unknowns need to be excluded, or they need to be assigned to all vehicle types considered based the proportions with known propulsion type. Judging by the results provided when the unknowns were excluded, this changes the conclusions, as EVs would have a significantly higher casualty rate than ICEVs.

I think the conclusion that EVs are at least as safe as ICEVs should really be revised to being that EVs with AVAS are at least as safe to more accurately reflect the results, and ensure the result is no misapplied to countries where AVAS is not required.

Following this and another reviewer's suggestion, we have now allocated the unassigned casualties to the different propulsion groups following the same distribution of the known ones. We have also used the more reliable VMT data from MoT tests as denominator (as per this reviewer's suggestion

later). As such our new estimates in Table 1 are different from before. However, the key conclusion – that EVs are not more dangerous than ICEVs remain valid.

We have removed RQ3 completely (but used it to partially explain the findings), and made changes accordingly in the revised version.

The crash data source is the STATS19 database of police reported crashes in the UK. I am unfamiliar with any particular limitations of this data, though I am aware that police reported crash data may have some coding errors, missing data, and for some biases, especially for variables that are self-reported by the driver. However, it is appropriate for use in this study, provided the problem posed by the missing data on vehicle propulsion is handled well.

As mentioned earlier, we have now addressed the missing propulsion issues in the revised version. Please note that we used the summary tables for our RQ1.

The paper refers to the data used in answer RQ2 as the “micro-dataset”. It is unclear what is source is, though I assume it is a subset of the STATS19 data. It is also unclear why the data was restricted to 2019-2022.

Table 1 used summary data from DfT– Table RAS0507 (which uses STATS19). This is clarified in revised version. The microdata used in Table 2 means STATS19 collision/casualty/vehicle record for every collision during that period. We have already mentioned that pre-2019 there were not enough EVs, 2022 was the last year microdata available when we started the work late 2023/early 2024.

The other main data used is vehicle miles travelled reported by the Department for Transport. I note that this data source could use an exact reference (e.g. was it from a report or series of reports?). The analysis assumes that equal vehicle miles are travelled by all vehicle types (EV, HEV, ICEV). This may or may not be true. I would like to see this study include some more supporting evidence (e.g. references) that support this assumption. There is a line in the Methods section, paragraph 4, that states “EVs reached parity with ICEVs in the UK in terms of annual driving per car in 2019”, but there is no reference. I note that different data on EV mileage is presented in the supporting information. Again, a clear reference and explanation of how that vehicle mileage was estimated would be useful. I think it is also a bit odd to have this as supporting information. If the author thinks the mileage data is superior, they should use it in the study.

We have now added clear references in the revised version, especially in the Methods section. We originally erred on the conservative side using DfT average mileage data for vehicles in the previous version. However, as per the reviewer’s suggestion, we recalculated the risk ratios using MOT test-derived VMT data as denominator for EVs (+ the reallocated missing propulsion casualties) and present them as the main results instead of relegating them to supporting information. All the necessary changes in the text have been made to reflect this. Please note that although the numbers have changed the main conclusions remain the same in the revised version.

I find the method used to answer RQ3 unusual. When considering a vehicle safety technology that was mandated from a certain point in time, the analysis would usually look at the year of manufacture of the vehicles, and look at those before and after. To use calendar year of crash is

unusual. This would mean that there would be many EVs and HEVs involved in pedestrian crashes in the Post-AVAS category that do not have AVAS. This problem is mitigated somewhat by the low number of EVs, and their rapid increase in sales. However, I don't see why the analysis used calendar year of crash rather than year of manufacture. Vehicle mileage could still be taken into account.

Our original logic was the same as the reviewer points out – that EV sales grew rapidly since 2019, so it was “more likely” that those vehicles will be fitted with AVAS, as their share goes up collisions would fall. However, we agree that this result may not be robust enough, as such we have removed RQ3 altogether in this version (although we use the results to partially explain other results).

I note that exact n and p values were not reported for all results, as is requested in the author guide.

*This is primarily related to Table 2. N has been reported in the model diagnostics near the end of the table. We used the *, **, *** identifiers describing statistical significance. In the revised version we present the p-values in supporting information Table S12.*

With regard to references, I found that there were many statements made or statistics quoted that lacked an appropriate reference to back them up. This includes.

Main-Paragraph 1, the first two sentences. *Added in the revised version*

Main-Paragraph 1, second to last sentence *added in the revised version*

Main – Paragraph 3, second to last sentence *added in the revised version*

Main, do pedestrians suffer more..., Paragraph 2, last sentence

This was our explanation of getting the statistically insignificant estimate. Did the reviewer want reference to the point that EVs are heavier? We have qualified the statement now, too.

If references to back up these statements cannot be provide, they should be deleted.

Reviewer #2 (Remarks to the Author):

NATCOM-Are electric vehicles more dangerous for pedestrians?

The study uses STATS19 to analyze pedestrian casualties related EVs, HEVs, and ICEVs. The findings show no significant difference in collision rates or injury severity between EVs and ICEVs. However, HEVs have a higher pedestrian casualty rate than ICEVs. The paper claims the introduction of AVAS in 2019 correlates with reduced casualty rates for both EVs and HEVs, suggesting policy effectiveness. The study concludes that EVs are at least as safe as ICEVs for pedestrians.

My key concerns include uncontrolled variables and insufficient separation of low-speed collisions to isolate AVAS impacts. In addition, AVAS policy analysis is questioned due to phased implementation (2019–2021) and the concurrent rollout of Autonomous Emergency Braking systems, potentially conflating causality.

In summary, I believe comments 7-9 must be addressed for publication.

We thank the reviewer for the valuable comments. Our response is in italics below, including our approach to address comments 7-9.

Major comments:

1. A few uncontrolled variables limit the extent of conclusions made from the available data. For example, it is implicitly assumed that the EV and ICEV drivers have the same driving skills. If the assumption is not true, it affects the conclusions. Considering the likely scenario of EVs being more attractive to younger drivers, the possible casualty rate difference may be due to this uncontrolled variable rather than the vehicle type. Pedestrian attitude toward vehicle type, pedestrian density, vehicle's physical condition, age, and mileage are other uncontrolled factors.

We have considered this in our Table 2, when we studied injury severity and used microdata. Table 1 uses DfTs summary data table RAS0507, which do not differentiate collisions by vehicle propulsion type and age. Even if we could develop these from microdata (a substantial task), this only provides us with the numerator of casualty rates in Table 1. It is still not possible to 'accurately' calculate casualty rates and then risk-ratios, as we would require 'exposure' (denominator) variable by each group (miles by young/old driver by different vehicle propulsion type + urban rural, as below), which is not available.

2. Another uncontrolled variable that affects the reliability of the conclusions is the accident location, urban or rural areas. The prevalence of EVs and ICEVs might differ in those areas, as well as the casualty rates and severity. For example, assume the casualty severities are lower in urban areas where EVs' presence is expected to be higher. Then, the tests will show that EVs have less severe accidents, when in fact, this is a result of the location rather than the vehicle type. Without separating the dataset, the results cannot be reliably attributed to vehicle type.

For injury 'severity' in Table 2, we have controlled for speed already, which is correlated with urban/rural classification. So this specific concern has already been accommodated in that model.

For casualty rates, as mentioned above, we used summary data for our Table 1, and urban/rural classification by propulsion type is not available. Also, even if we could separate urban and rural 'collisions', the denominator – mileage by different propulsion types and urban/rural types – is not available. Even if we go back to microdata, exposure (mileage) will remain an important variable and is not available at the required resolution, as above.

We agree with the reviewer's proposition that urban/rural may have an important role in casualty rates, and we have already acknowledged that in the original version. We clarify this issue in general further in the revised version discussion when we said "As such, the issue may be less a result of whether the vehicles are HEV or ICEV, rather where and how the vehicles are driven. Investigating such potential confounding factors is an important avenue for future research – both for HEVs and EVs".

Please note that we could have modelled the strength of association of these variables – age, urban rural location etc. – with collisions, but that would have been conditioned on a collision happening and cannot be used to calculate casualty 'rates' (accurate exposure/denominator not available), which is our main interest in RQ1/Table 1.

3. "We could not model urban and rural areas separately, given speed limits are highly correlated with urban/rural indicator."

Since the database contains the location information of the accidents, I wonder why it was impossible to model urban and rural areas separately. Without this separation, it is difficult to attribute results reliably to vehicle type.

This refers to injury severity model using micro data. As the reviewer points out correctly, we have information about urban and rural indicator in the micro dataset. However, that variable is high correlated with speed limits (rural areas will have higher speed limits generally). We can include these, but it is well-known that such multicollinearity affects parameter estimates and inference substantially. We had indeed tested for speed only, urban-rural only and speed and urban/rural both in the model, and the speed only model was the best model using model fit statistics. We explained it in the Methods section of the revised version.

4. In Table 1, all the rows are compared to row 1, 2014. As mentioned in the text, due to the fluctuations and deficiencies in EVs' early years, this is not a proper base. I suggest using the year 2018 or 2019. The same applies to Table 3.

The rows are actually not compared to row 1 in Table 1. The rate ratios (along with statistical significance) are calculated within the relevant pairs each year (each row) or multiple years (last row).

5. In Table 1, most rows are different from the base, but we don't know if they are different from each other. In addition, I suggest using notched boxplots for visual statistical significance tests between the cases that are discussed or built upon in the manuscript.

As mentioned above, the differences and statistical significances are not for between-row comparisons, but for between relevant columns within each row. These estimates are based on point summary statistics of DfT's Table RAS0507, so it is not feasible to plot box plots.

6. Like comment 4, in Table 2, the statistical significance test bases are apparently not properly chosen, for example, day-time combination or speed limit. Moreover, sample sizes are not mentioned. I believe notched box plots with sample sizes would resolve the issue. It will be much easier to read and interpret the significance between every pair. I recommend adding it to the supplementary material.

Unlike Table 1, Table 2 results are direct parameter estimates of a logistic regression. We were not comparing simple means of different groups, for which notched box plot would have been useful. Sample size for the logit regression has already been reported at the end of Table 2. We do not believe there is any mistake in the choice of 'base' levels in our regression model – for day time combination, weekday-morning peak is base, for speed limit 20 mph is base. Please note that this choice does not alter any of the results, the parameters will simply be re-adjusted for another base level, but the conclusions regarding the levels remain the same.

7. Regarding the mandated Acoustic Vehicle Alerting System, in July 2019 it was implemented on only “new models”. From July 2021, all “newly registered” models must have the alarm. Therefore, a large portion of EVs manufactured before 2019, and old models produced between 2019 and 2021 are not required to have the AVAS. As expected, the policy would take a couple of years to be fully implemented. I believe the casualty rate drop is unlikely to be related to the new policy, as the majority were not equipped with the system in the years studied in the paper, e.g., 2019-2023.

We are afraid that we do not fully agree with the reviewer here: not only did the EV sales grow rapidly during the 2019-2023 period (Fig 1b and c), the number of available new EV models have also increased rapidly during this period. This means that there were likely a large share of 2019-2023 EVs on road were fitted with AVAS. Please note that we have removed RQ3 as a key research question in the revised version, although we keep the result to partially explain other results.

8. EVs are required to emit sound at speeds lower than 20 km/h. At higher speeds, the policy is not mandated. Therefore, to study the effects of the policy, only casualties at low speed should be compared. However, the manuscript compares all casualties before and after the implementation at all speed ranges; see Table 3. If the effect is significant only at low speeds, it is evidence of the new regulation's effectiveness. But if the effect is significant at all speed ranges, then, it is evidence against it being related to the policy. The paper doesn't separate the low-speed cases and therefore the statistical significance test doesn't support the policy's effectiveness.

We agree that investigating low speed collisions only would have been a more robust approach. However, DfT summary statistics do not differentiate collisions segregated by speed, vehicle type and propulsion types. Please note that we would have needed VMT for these groups for the denominator, too. We have clearly mentioned this as a future avenue of research in the original and revised version of the paper.

9. Autonomous Emergency Braking (AEB) with pedestrian detection was introduced two or three years before AVAS. The technology had a couple of years to show its effectiveness during the period studied in the paper. It is not speed-related and is compatible with the results. The casualty drops may be related to the technology upgrade.

We have already mentioned the possibility of EVs having better collision avoidance technology. AEB in the UK, however, was mandated from 2022 for new vehicle models, and only from 2024 for all new vehicles sold. So that specific policy has less of a direct role in this context. The technology was getting introduced in high-end vehicles (and some EVs too), and we have clearly acknowledged that in the original and revised version of the paper.

In response to comments 7-9, we have reframed the paper, and removed RQ3, as we acknowledge that that finding is not robust yet. We use some of that analysis in partially explaining the findings.

10. The study has a multiple comparisons problem. When performing many statistical significance pair-wise tests, for example in Table 2, the chance of getting accidental statistical significance increases. I suggest performing a false discovery rate analysis like Benjamini-Hochberg.

Multiple comparison problem arises when the primary objective is drawing inference on a large number of variables. Our key variable of interest in the regression is the role of EV dummy, and the rest of the variables are there to control for other factors and less important in the context of our research question. Nonetheless, we did draw some inferences for those too, and agree with the reviewer that we need to correct the confidence intervals for multiple comparison. We have now applied the Benjamin-Hochberg correction as per the reviewer's suggestion, the core results do not change. Only 5 of the ancillary variables change significance (out of ~40).

Minor comments:

11. Since the novelty of the study is the investigation of EVs, which were not present in the literature before, I suggest adding a few lines explaining the motivation. I am looking for reasons why the authors thought EVs may be different from HEVs.

We believe we have mentioned that in the introduction, quietness and heavier weight of EVs?

12. "Even those studies that exist sometimes report contradictory findings (Edward et al. 2024, Morgan et al. 2011)."

I don't see any contradiction between the two studies. Edward et al. discuss that the number of accidents per travel length is higher for EVs, while Morgan et al. focus on visually impaired pedestrians and the increased collision risk with quiet vehicles.

While Morgan et al.'s primary focus was indeed visually impaired pedestrians, they did do a comparison using STATS19 dataset before getting into their main focus. Although they did not conduct any statistical tests, they did mention "relative to the number of registered vehicles, E/HE vehicles were 10% less likely to be involved in a collision with a pedestrian than ICE vehicles".

Edwards et al. found that there are higher risks associated with EV/HEVs. We have rephrased to say “.. studies that exist, sometimes do not agree ..”.

13. In Fig. 1, there are a few issues; it is unclear what walk-miles and ICEV-Low are (ICEV-Low is defined in the methods section. Still, I suggest defining them in captions); all the y-axes' titles are vague or partially described; the x-axis in (a) is truncated.

ICEV now has only one estimate after the suggested revisions are undertaken. The figures are therefore all new, and are legible (pending any issues during submission).

Walk miles are miles walked by pedestrians, similar to vehicle miles driven by vehicles. We did not use it in calculations, but believe it is an interesting statistics.

14. When using casualty rate and aggregated casualty rate, please explicitly define what you mean.

Casualty rates in Table 1 and 3 use summary statistics (i.e. not derived by us from microdata as in Table 2), we have removed 'aggregated' from this version as it could lead to confusion.

15. Please describe what each data point is (I guess a recorded accident?), and mention the size of the groups being statistically compared or how many samples each group has.

For Table 2, which uses microdata, each data point is indeed represent a casualty record. The sample size (53,090) is clearly mentioned in diagnostic statistics near the end of Table 2. For Table 1, the estimates are single point estimates directly from Table0507 of DfT. This is clarified further both in the main text and Methods.

16. In the text, HEVs are more dangerous than EVs and ICEVs. Since this is unintuitive, it would be helpful if the authors explained the reason.

We have attempted a few potential reasons for why HEV casualty rates could be high (HEVs are very popular as taxis/PHVs in the UK, so they are driven more and are driven in urban areas more, so closer to pedestrians).

17. In Table 1, please define the abbreviations in the caption for readability. Moreover, please mention the number of samples in each group.

As mentioned above, these are summary statistics used directly from DfT's Table0507, so sample size is not available. The abbreviations – EV, HEV, ICEV – have been described earlier in the text?

18. “RQ2 is answered using the 'casualty' micro-dataset for 2019-2022.”

Why did you exclude 2023 for RQ2, whereas it was used for RQ1?

2023 microdata became available after that part of the analysis was completed. There is also no reason to believe that the underlying factors may have changed in 2023. It appears from the timeseries data that EV collisions had already stabilised, too. The vehicle model data (especially SUV

identification) was available to us only until 2022. For RQ1, which uses summary data, more years were available.

19. After Table 1, it would be helpful to explain how the binary logit regression model is applied. *We have explained it in more detail in Methods section. [this is a challenge for Nature papers, where the Methods comes after the main paper].*

20. I think the speed limit for AVAS is 20 km/h and not 20 mph.

Indeed. We have corrected the error (12 mph).

21. The authors mention

“As a sensitivity analysis, we removed the missing-propulsion type casualties from ICEVs and re-estimated the ICEV casualty rates (ICEV-Low in Fig. 1 and Table 1). Even in this extreme and highly unlikely scenario, EV casualty rates are statistically not different from ICEV rates for five of the ten years (Table 1).”

And

“We also run a lower estimate for ICEV casualty rate (ICEV-low), where we do not include the unknown collisions to ICEVs, but believe it grossly underestimates ICEV casualty rates.”

The exclusion makes perfect sense. Why do you think it is highly unlikely, and it grossly underestimates casualty rates? Aren't the excluded data supposed to have the same distribution characteristics as the whole set?

It would grossly underestimate the 'rate' for ICEVs because the denominator includes 'all' ICEV miles. We have now redistributed the 'unknown' category assuming the same distribution in the 'unknown' category, as the reviewer alluded to. We have also moved the supporting info Table 1 to main results where we used the 'more' accurate distribution of EV miles from MoT data, as per another reviewer's suggestion (and adjusted the other miles accordingly). As such the exact numbers in this revised version now differ from the original submission, however, the key conclusions remain the same.

22. In the car body type study, please explain the dummy variable and how you use it.

It is 'one' regression model (not a separate 'car body type' study). The dummy/categorical variable for SUV enters as an explanatory factor (=1 if it is a large SUV, =0 if it is not, base in results is 0, ie not SUV) in the regression.

23. The abstract mentions quieter driving and heavier weight as the reasons for EV safety concerns. The AVAS study justifies the former but I don't see a reason to mention the latter in the abstract. The introduction may discuss heavier weight and its implications with proper citations.

We believe we mentioned heavy weight (along with citation) in the introduction section in the original (and this) version. Note that we have reframed the paper and removed RQ3 (impact of AVAS) in this version. So AVAs does not appear in abstract anymore. We mentioned weight in abstract because accident severity indeed can depend on vehicle weight and EVs are heavier than similar ICEVs.

24. Why are the authors talking about car body type and SUVs? I don't see how they relate to the paper's research questions. Can you explain why you are interested in SUVs or separating them from the rest?

We are not necessarily interested in SUVs per se, but SUVs are generally heavier vehicles and also more dangerous (i.e. they impart more severe injury) to pedestrians due to their body shape (higher bonnet height). A large share of early EVs for most manufacturers were SUVs too [e.g. Tesla Model Y, Jaguar IPace, Volvo XC Recharge, Toyota BZ40, MGZS EV, Kia Soul EV etc.]. As such, it is important to control for that body-shape in the regression. Note that the regression is not for casualty risk, but for injury severity when a collision happens.

We thank the reviewer for the detailed comment.

Reviewer #3 (Remarks to the Author):

The manuscript addresses an important research question regarding the pedestrian safety implications of electric vehicles. While the topic is timely and the use of collision data from STATS19 is appropriate, several methodological concerns limit the reliability of the findings. The following issues should be addressed in a future iteration of the paper.

We thank the reviewer for the positive view of the paper. Our responses to the reviewer's comments are given below in italics.

1) It is not clear how the exposure data were obtained and processed. The STATS19 database includes only collision records and provides very limited traffic volume or exposure information. The paper should clearly explain the source of traffic data and the method used to allocate exposure across vehicle propulsion types.

We have now described how we get the exposure (VMT) data in detail in the Methods section of the revised version.

2) Similarly, the methodology for estimating annual miles driven by different vehicle types is inadequately described. The paper offers little information on how these figures were derived or adjusted for differences in usage patterns (e.g., taxis vs. private vehicles), which is essential for accurate rate calculations.

VMT is the exposure data. This is now described in the revised version. We did not separate taxis vs cars, they were combined for Table 1. For Table 2, mileage data was not used.

3) The binary logit regression model used for injury severity includes a large number of statistically insignificant variables. This overfitting may introduce bias and reduce the reliability of the estimates. A more parsimonious approach to model specification with only statistically significant is recommended.

We are afraid we disagree with this approach. While in some fields (e.g. choice modelling) this is a common approach, for secondary data analysis with a decent volume of data, it is quite common to keep statistically insignificant parameters, as they tell a story, too. Please note that the variables themselves are not statistically insignificant, but a few of the 'levels' describing a variable were insignificant. We have now adjusted the confidence intervals for multiple comparison (many variables) in the revised version following another reviewer comment. This makes our findings more robust.

4) The use of the Poisson distribution for computing confidence intervals may be problematic, as collision data are often over-dispersed. The Poisson model assumes the mean equals the variance, which is rarely the case in traffic safety data. Alternative methods should be considered.

Our main conclusions are based on the 5 year post data (2019-2023), for which we have now estimated both Negative Binomial (which accounts for overdispersion) and Poisson based models. As

the reviewer alluded to, the Negative binomial indeed was the better model. Our main conclusion remain the same (that casualty rate ratios for EVs are statistically not different from ICEVs) since NB has a wider confidence interval than Poisson. The results are presented in Table 1 and Table 3 in the revised version.

For individual year pair-wise comparison, it is not possible to estimate the overdispersion parameter in NB model (only one summary data point for each for each propulsion type). As such, for individual years, we keep Poisson results (please note that we do not make conclusions on this basis). This is described mentioned in the Methods section in the revised version.

5) The use of a 10-year dataset without accounting for potential structural changes over time raises concerns. Multi-year data can exhibit temporal instability, where parameter relationships shift due to evolving technologies, policies, or external factors (e.g., COVID-19). This issue is well-documented in the literature—see, for example, Mannering (2018), “Temporal instability and the analysis of highway accident data.” Addressing this concern may require segmented analysis or the inclusion of interaction terms to capture time-based variation.

We have now removed RQ3 from the revised version, although we keep some of that analysis in our reframed paper to explain some changes. Also, please note that we did present a counterfactual on how ICEV casualty rates evolved in the original version, which tracks temporal changes that may have affected both. We also did not run any regression (for original RQ3) which could have used an interaction term.

6) The literature review omits key studies focusing on safety issues of similar types of vehicles.

We believe we had done a comprehensive literature search on the safety implications of EVs that uses actual collision data and have mentioned the relevant studies in either introduction or discussion section (Nature paper structure does not allow ‘typical’ literature review section). We will be glad if the reviewer could point us to the key EV collision related studies that we may have missed.

Response to Editor's and Reviewers' Comments

Our responses are in italics.

Reviewer #1 (Remarks to the Author):

I believe all my comments have been adequately addressed. Thank you.

Thank you for your suggestions, and especially for a quick review.

Reviewer #2 (Remarks to the Author):

We respond below only to the unresolved comments, and deleted those that the reviewer was happy with.

1. I see. Then, I recommend adding a statement to the limitation section mentioning that the insufficient partitioning in the source data prevented the authors from isolating such effects. I like the reader to know there are other untested hypotheses that may change the conclusions if better portioned data becomes available.

We have mentioned this in the revised version discussion section now – “As such, the issue may be less a result of whether the vehicles are HEV or ICEV, rather where and how the vehicles are driven. Investigating such potential confounding factors (e.g. driver age or pedestrian density of areas where the vehicles are driven) is an important avenue for future research – both for HEVs and EVs. Obtaining exposure data for such detailed analysis is a potential challenge.”

4. I see. Please add a note, maybe in the table caption to avoid confusion.

We have added column no and used mathematical notation on which column is divided by which one to get the ratios in revised Table 1. We hope this will be avoid any confusion now.

5. Same as 4.

We do not think 5 is applicable now in light of changes in 4 above.

6. I still think a different base selection better represents the data. However, as the authors mention, it doesn't change the results. I leave it to the authors' judgment.

We would prefer to stick our baseline – for most variables we think our chosen baselines are easier to interpret (e.g. from 20 mph speed limit base, each successive speed limits are progressively worse).

7. The statement requires supporting data. Any further build-up on the assumption of the majority of EVs having AVAS from 2019 to 2023 must be carefully examined (and what majority, 60%? 70%? 95%?). I agree that as time passes, the percentage increases, but is it fast enough for the effect to be visible in casualty rates? I still believe that this is not the case.

Overall, I recommend a year-wise percentage study for EVs with and without AVAS.

If impossible, please add a note regarding the rollout speed limitation, as I explained in the original comment where the authors are discussing AVAS effectiveness. Please clearly mention both views and the reason why you believe it is not the case.

(Currently, the authors ignore the point and just mention that the result is not robust due to other reasons.)

As mentioned in the last version, RQ3 - the effectiveness of the AVAS introduction on reducing casualty rates - had been dropped from the revised and this version of the paper. As such it is not clear why the paper needs further data support/analysis to prove this (the reviewer also agrees that the percentages increase as time passes – which was our point).

EV numbers jumped from only 50,000 at the end of 2018 to 292,000 at the end of 2021 to 806,000 at the end of 2023, indicating that at least around two-thirds ((806-292)/806) of the EVs had AVAS fitted by 2023. This is accompanied by a rapid increase in the number of new vehicle models from 2019 (and updates of older models, which also require type approval, so should have had AVAS). On top of it, some early EVs had AVAS even before 2019 – e.g. Nissan Leaf, BMW i3 (optional), Jaguar iPACE – all are pre-2019 cars, but had AVAS. All of these suggest there were rapidly increasing and adequate number of AVAS equipped EVs post 2019.

Although it is no longer a RQ, we have modified the discussion to the following in the revised version:

“... EV numbers jumped from 292,000 at the end of 2021 to 806,000 at the end of 2023, accompanied by a rapidly increasing number of new EV models since 2019 (all new models required AVAS), which suggests that a large share of EVs had AVAS post 2019. These together indicate that the AVAS regulation likely contributed to reduce pedestrian casualty rates of EVs and HEVs. While robust conclusions cannot be drawn here, future studies should investigate changes in casualty rates by speed categories when relevant data become available: a larger fall in low-speed causalities for EVs compared to ICEVs will lend further support to a causal association.”

8. I understand that the data is limited. But I believe that isolating low speeds is necessary for the conclusions made in the original submission. In the current version, a more direct mention of speed study is helpful (the current form is too vague).

We directly mentioned the need for ‘speed study’ in the last revised and this version (before Table 3, which is now in Discussion section)-

“While robust conclusions cannot be drawn here, future studies should investigate changes in casualty rates by speed categories when relevant data become available: a larger fall in low-speed causalities for EVs compared to ICEVs will lend further support to a causal association.”

Please note that we had removed RQ3 in the revised version altogether – so we are not drawing any conclusions about the effectiveness of the AVAS policy (which was RQ3) in the most recent versions.

11. Yes. But how are those important in the analysis?

We have now explicitly added in the introduction section that EVs can be heavier than ICEVs and HEVs and that the driving patterns could be different, too.

13. Please make sure that the terms are properly introduced. Thankyou.

They have now been.

16. I assume it is already explained in the text.

Yes, the explanation was added in the last version, and remains in this version too (after Table 3 in discussion section).

17. I know. I meant for readers who look at the figures and tables first. Please explain that the sample size is not available and why, where it fits in the manuscript, if not already explained.

We have explained the abbreviations on figures and tables in the revised version now.

18. I see. Please explain it in the text for the curious reader.

Explained in the revised version Methods section.

We thank the reviewer for their thoughtful comments, and super quick 2nd review.